# MPNet: Masked and Permuted Pre-training for Language Understanding

**Kaitao Song[1], Xu Tan[2], Tao Qin[2], Jianfeng Lu[1], Tie-Yan Liu[2]**
[1]Nanjing University of Science and Technology, [2]Microsoft Research
{kt.song,lujf}@njust.edu.cn, {xuta,taoqin,tyliu}@microsoft.com

## Abstract

BERT adopts masked language modeling (MLM) for pre-training and is one of the most successful pre-training models. Since BERT neglects dependency among predicted tokens, XLNet introduces permuted language modeling (PLM) for pre-training to address this problem. However, XLNet does not leverage the full position information of a sentence and thus suffers from position discrepancy between pre-training and fine-tuning. In this paper, we propose MPNet, a novel pre-training method that inherits the advantages of BERT and XLNet and avoids their limitations. MPNet leverages the dependency among predicted tokens through permuted language modeling (vs. MLM in BERT), and takes auxiliary position information as input to make the model see a full sentence and thus reducing the position discrepancy (vs. PLM in XLNet). We pre-train MPNet on a large-scale dataset (over 160GB text corpora) and fine-tune on a variety of down-streaming tasks (GLUE, SQuAD, etc). Experimental results show that MPNet outperforms MLM and PLM by a large margin, and achieves better results on these tasks compared with previous state-of-the-art pre-trained methods (e.g., BERT, XLNet, RoBERTa) under the same model setting. The code and the pre-trained models are available at: https://github.com/microsoft/MPNet.

## 1 Introduction

Pre-training language models [1, 2, 3, 4, 5, 6, 7, 8] have greatly boosted the accuracy of NLP tasks in the past years. One of the most successful models is BERT [2], which mainly adopts masked language modeling (MLM) for pre-training[1]. MLM leverages bidirectional context of masked tokens efficiently, but ignores the dependency among the masked (and to be predicted) tokens [5].

To improve BERT, XLNet [5] introduces permuted language modeling (PLM) for pre-training to capture the dependency among the predicted tokens. However, PLM has its own limitation: Each token can only see its preceding tokens in a permuted sequence but does not know the position information of the full sentence (e.g., the position information of future tokens in the permuted sentence) during the autoregressive pre-training, which brings discrepancy between pre-training and fine-tuning. Note that the position information of all the tokens in a sentence is available to BERT while predicting a masked token.

In this paper, we find that MLM and PLM can be unified in one view, which splits the tokens in a sequence into non-predicted and predicted parts. Under this unified view, we propose a new pre-training method, masked and permuted language modeling (MPNet for short), which addresses the issues in both MLM and PLM while inherits their advantages: 1) It takes the dependency among the predicted tokens into consideration through permuted language modeling and thus avoids the

issue of BERT; 2) It takes position information of all tokens as input to make the model see the position information of all the tokens and thus alleviates the position discrepancy of XLNet.

We pre-train MPNet on a large-scale text corpora (over 160GB data) following the practice in [5, 7], and fine-tune on a variety of down-streaming benchmark tasks, including GLUE, SQuAD, RACE and IMDB. Experimental results show that MPNet outperforms MLM and PLM by a large margin, which demonstrates that 1) the effectiveness of modeling the dependency among the predicted tokens (MPNet vs. MLM), and 2) the importance of the position information of the full sentence (MPNet vs. PLM). Moreover, MPNet outperforms previous well-known models BERT, XLNet and RoBERTa by 4.8, 3.4 and 1.5 points respectively on GLUE dev sets under the same model setting, indicating the great potential of MPNet for language understanding.

## 2   MPNet

### 2.1   Background

The key of pre-training methods [1, 2, 4, 5, 10] is the design of self-supervised tasks/objectives for model training to exploit large language corpora for language understanding and generation. For language understanding, masked language modeling (MLM) in BERT [2] and permuted language modeling (PLM) in XLNet [5] are two representative objectives. In this section, we briefly review MLM and PLM, and discuss their pros and cons.

**MLM in BERT**   BERT [2] is one of the most successful pre-training models for natural language understanding. It adopts Transformer [11] as the feature extractor and introduces masked language model (MLM) and next sentence prediction as training objectives to learn bidirectional representations. Specifically, for a given sentence $x = (x_1, x_2, \cdots, x_n)$, MLM randomly masks $15\%$ tokens and replace them with a special symbol $[M]$. Denote $\mathcal{K}$ as the set of masked positions, $x_{\mathcal{K}}$ as the set of masked tokens, and $x_{\backslash \mathcal{K}}$ as the sentence after masking. As shown in the example in the left side of Figure 1(a), $\mathcal{K} = \{2, 4\}$, $x_{\mathcal{K}} = \{x_2, x_4\}$ and $x_{\backslash \mathcal{K}} = (x_1, [M], x_3, [M], x_5)$. MLM pre-trains the model $\theta$ by maximizing the following objective

$$\log P(x_{\mathcal{K}}|x_{\backslash \mathcal{K}}; \theta) \approx \sum_{k \in \mathcal{K}} \log P(x_k|x_{\backslash \mathcal{K}}; \theta). \tag{1}$$

**PLM in XLNet**   Permuted language model (PLM) is proposed in XLNet [5] to retain the benefits of autoregressive modeling and also allow models to capture bidirectional context. For a given sentence $x = (x_1, x_2, \cdots, x_n)$ with length of $n$, there are $n!$ possible permutations. Denote $\mathcal{Z}_n$ as the permutations of set $\{1, 2, \cdots, n\}$. For a permutation $z \in \mathcal{Z}_n$, denote $z_t$ as the $t$-th element in $z$ and $z_{<t}$ as the first $t - 1$ elements in $z$. As shown in the example in the right side of Figure 1(b), $z = (1, 3, 5, 2, 4)$, and if $t = 4$, then $z_t = 2$, $x_{z_t} = x_2$ and $z_{<t} = \{1, 3, 5\}$. PLM pre-trains the model $\theta$ by maximizing the following objective

$$\log P(x; \theta) = \mathbb{E}_{z \in \mathcal{Z}_n} \sum_{t=c+1}^{n} \log P(x_{z_t}|x_{z_{<t}}; \theta), \tag{2}$$

where $c$ denotes the number of non-predicted tokens $x_{z_{<=c}}$. In practice, only a part of last tokens $x_{z_{>c}}$ (usually $c = 85\% * n$) are chosen to predict and the remaining tokens are used as condition in order to reduce the optimization difficulty [5].

**Pros and Cons of MLM and PLM**   We compare MLM and PLM from two perspectives: the dependency in the predicted (output) tokens and the consistency between pre-training and fine-tuning in the input sentence.

- **Output Dependency**: As shown in Equation 1, MLM assumes the masked tokens are independent with each other and predicts them separately, which is not sufficient to model the complicated context dependency in natural language [5]. In contrast, PLM factorizes the predicted tokens with the product rule in any permuted order, as shown in Equation 2, which avoids the independence assumption in MLM and can better model dependency among predicted tokens.

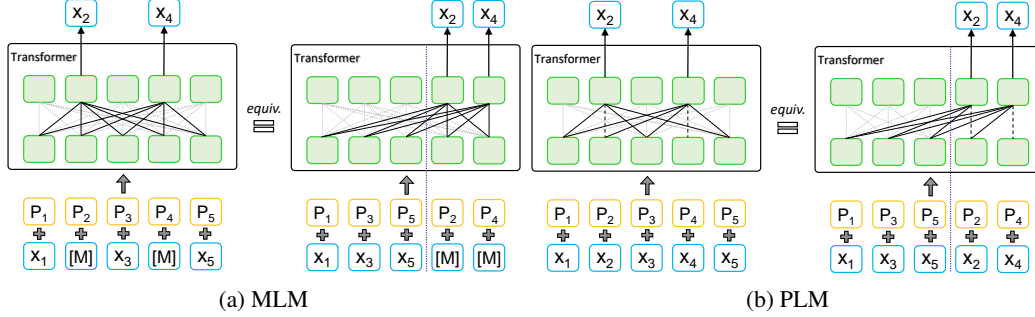

(a) MLM                                                      (b) PLM

Figure 1: A unified view of MLM and PLM, where $x_i$ and $p_i$ represent token and position embeddings. The left side in both MLM (a) and PLM (b) are in original order, while the right side in both MLM (a) and PLM (b) are in permuted order and are regarded as the unified view.

- **Input Consistency** Since in fine-tuning of downstream tasks, a model can see the full input sentence, to ensure the consistency between pre-training and fine-tuning, the model should see as much information as possible of the full sentence during pre-training. In MLM, although some tokens are masked, their position information (i.e., the position embeddings) are available to the model to (partially) represent the information of full sentence (how many tokens in a sentence, i.e., the sentence length). However, each predicted token in PLM can only see its preceding tokens in a permuted sentence but does not know the position information of the full sentence during the autoregressive pre-training, which brings discrepancy between pre-training and fine-tuning.

As can be seen, PLM is better than MLM in terms of leveraging output dependency while worse in terms of pre-training and fine-tuning consistency. A natural question then arises: can we address the issues in both MLM and PLM while inherit their advantages?

## 2.2 A Unified View of MLM and PLM

To address the issues and inherit the advantages of MLM and PLM, in this section, we provide a unified view to understand MLM and PLM. Both BERT and XLNet take Transformer [11] as their backbone. Transformer takes tokens and their positions as input, and is not sensitive to the absolute input order of those tokens, only if each token is associated with its correct position in the sentence.

This inspires us to propose a unified view for MLM and PLM, which rearranges and splits the tokens into non-predicted and predicted parts, as illustrated in Figure 1. For MLM in Figure 1(a), the input in the left side is equal to first permuting the sequence and then masking the tokens in rightmost ($x_2$ and $x_4$ are masked in the permuted sequence $(x_1, x_3, x_5, x_2, x_4)$ as shown in the right side). For PLM in Figure 1(b), the sequence $(x_1, x_2, x_3, x_4, x_5)$ is first permuted into $(x_1, x_3, x_5, x_2, x_4)$ and then the rightmost tokens $x_2$ and $x_4$ are chosen as the predicted tokens as shown in the right side, which equals to the left side. That is, in this unified view, the non-masked tokens are put on the left side while the masked and to be predicted tokens are on the right side of the permuted sequence for both MLM and PLM.

Under this unified view, we can rewrite the objective of MLM in Equation 1 as

$$\mathbb{E}_{z \in \mathcal{Z}_n} \sum_{t=c+1}^{n} \log P(x_{z_t} | x_{z_{<=c}}, M_{z_{>c}}; \theta), \qquad (3)$$

where $M_{z_{>c}}$ denote the mask tokens [M] in position $z_{>c}$. As shown in the example in Figure 1(a), $n = 5$, $c = 3$, $x_{z_{<=c}} = (x_1, x_3, x_5)$, $x_{z_{>c}} = (x_2, x_4)$ and $M_{z_{>c}}$ are two mask tokens in position $z_4 = 2$ and $z_5 = 4$. We also put the objective of PLM from Equation 2 here

$$\mathbb{E}_{z \in \mathcal{Z}_n} \sum_{t=c+1}^{n} \log P(x_{z_t} | x_{z_{<t}}; \theta). \qquad (4)$$

From Equation 3 and 4, under this unified view, we find that MLM and PLM share similar mathematical formulation but with slight difference in the conditional part in $P(x_{z_t} | \cdot; \theta)$: MLM conditions on $x_{z_{<=c}}$ and $M_{z_{>c}}$, and PLM conditions on $x_{z_{<t}}$. In the next subsection, we describe how to modify the conditional part to address the issues and inherit the advantages of MLM and PLM.

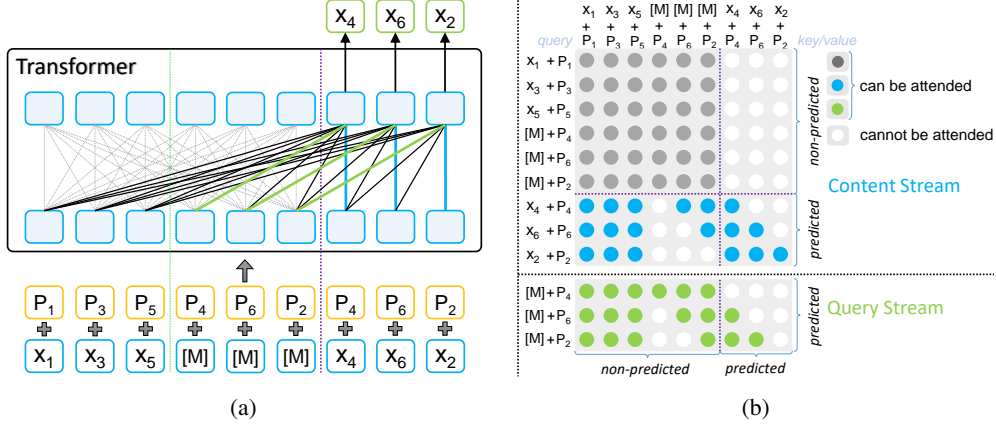

<div align="center">(a)                               (b)</div>

Figure 2: (a) The structure of MPNet. (b) The attention mask of MPNet. The light grey lines in (a) represent the bidirectional self-attention in the non-predicted part $(x_{z_{\leq=c}}, M_{z_{>c}}) = (x_1, x_5, x_3, [M], [M], [M])$, which correspond to the light grey attention mask in (b). The blue and green mask in (b) represent the attention mask in content and query streams in two-stream self-attention, which correspond to the blue, green and black lines in (a). Since some attention masks in content and query stream are overlapped, we use black lines to denote them in (a). Each row in (b) represents the attention mask for a query position and each column represents a key/value position. The predicted part $x_{z_{>c}} = (x_4, x_6, x_2)$ is predicted by the query stream.

## 2.3  Our Proposed Method

Figure 2 illustrates the key idea of our proposed MPNet. The training objective of MPNet is

$$\mathbb{E}_{z\in\mathcal{Z}_n} \sum_{t=c+1}^{n} \log P(x_{z_t}|x_{z_{<t}}, M_{z_{>c}}; \theta). \tag{5}$$

As can be seen, MPNet conditions on $x_{z_{<t}}$ (the tokens preceding the current predicted token $x_{z_t}$) rather than only the non-predicted tokens $x_{z_{\leq=c}}$ in MLM as shown in Equation 3; comparing with PLM as shown in Equation 4, MPNet takes more information (i.e., the mask symbol $[M]$ in position $z_{>c}$) as inputs. Although the objective seems simple, it is challenging to implement the model efficiently. To this end, we describe several key designs of MPNet in the following paragraphs.

**Input Tokens and Positions**  We illustrate the input tokens and positions of MPNet with an example. For a token sequence $x = (x_1, x_2, \cdots, x_6)$ with length $n = 6$, we randomly permute the sequence and get a permuted order $z = (1, 3, 5, 4, 6, 2)$ and a permuted sequence $x_z = (x_1, x_3, x_5, x_4, x_6, x_2)$, where the length of the non-predicted part is $c = 3$, the non-predicted part is $x_{z_{\leq=c}} = (x_1, x_3, x_5)$, and the predicted part is $x_{z_{>c}} = (x_4, x_6, x_2)$. Additionally, we add mask tokens $M_{z_{>c}}$ right before the predicted part, and obtain the new input tokens $(x_{z_{\leq=c}}, M_{z_{>c}}, x_{z_{>c}}) = (x_1, x_3, x_5, [M], [M], [M], x_4, x_6, x_2)$ and the corresponding position sequence $(z_{\leq=c}, z_{>c}, z_{>c}) = (p_1, p_3, p_5, p_4, p_6, p_2, p_4, p_6, p_2)$, as shown in Figure 2a. In MPNet, $(x_{z_{\leq=c}}, M_{z_{>c}}) = (x_1, x_3, x_5, [M], [M], [M])$ are taken as the non-predicted part, and $x_{z_{>c}} = (x_4, x_6, x_2)$ are taken as the predicted part. For the non-predicted part $(x_{z_{\leq=c}}, M_{z_{>c}})$, we use bidirectional modeling [2] to extract the representations, which is illustrated as the light grey lines in Figure 2a. We will describe how to model the dependency among the predicted part in next paragraph.

**Modeling Output Dependency with Two-Stream Self-Attention**  For the predicted part $x_{z_{>c}}$, since the tokens are in the permuted order, the next predicted token could occur in any position, which makes it difficult for normal autoregressive prediction. To this end, we follow PLM to adopt two-stream self-attention [5] to autoregressively predict the tokens, which is illustrated in Figure 3. In two-stream self-attention, the query stream can only see the previous tokens and positions as well as current position but cannot see the current token, while the content stream can see all the previous and current tokens and positions, as shown in Figure 2a. For more details about two-stream self-attention, please refer to [5]. One drawback of two-stream self-attention in PLM is that it can only see the

<div align="center">4</div>

previous tokens in the permuted sequence, but does not know the position information of the full sentence during the autoregressive pre-training, which brings discrepancy between pre-training and fine-tuning. To address this limitation, we modify it with position compensation as described next.

**Reducing Input Inconsistency with Position Compensation**   We propose position compensation to ensure the model can see the full sentence, which is more consistent with downstream tasks. As shown in Figure 2b, we carefully design the attention masks for the query and content stream to ensure each step can always see $n$ tokens, where $n$ is the length of original sequence (in the above example, $n = 6$)[2]. For example, when predicting token $x_{z_5} = x_6$, the query stream in the original two-stream attention [5] takes mask token $M_{z_5} = [M]$ and position $p_{z_5} = p_6$ as the attention query, and can only see previous tokens $x_{z_{<5}} = (x_1, x_3, x_5, x_4)$ and positions $p_{z_{<5}} = (p_1, p_3, p_5, p_4)$ in the content stream, but cannot see positions $p_{z_{>=5}} = (p_6, p_2)$ and

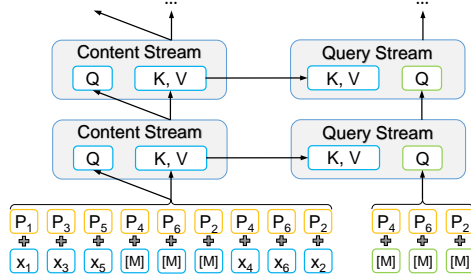

Figure 3: The two-stream self-attention mechanism used in MPNet, where the query stream reuses the hidden from the content stream to compute key and value.

thus miss the full-sentence information. Based on our position compensation, as shown in the second last line of the query stream in Figure 2b, the query stream can see additional tokens $M_{z>=5} = ([M], [M])$ and positions $p_{z_{>=5}} = (p_6, p_2)$. The position compensation in the content stream follows the similar idea as shown in Figure 2b. In this way, we can greatly reduce the input inconsistency between pre-training and fine-tuning.

| Model | Factorization |
|---|---|
| MLM | $\log P(\text{sentence} \mid \text{the task is } [M]\ [M]) + \log P(\text{classification} \mid \text{the task is } [M]\ [M])$ |
| PLM | $\log P(\text{sentence} \mid \text{the task is}) \qquad\quad + \log P(\text{classification} \mid \text{the task is sentence})$ |
| MPNet | $\log P(\text{sentence} \mid \text{the task is } [M]\ [M]) + \log P(\text{classification} \mid \text{the task is sentence } [M])$ |

Table 1: An example sentence "the task is sentence classification" to illustrate the conditional information of MLM, PLM and MPNet.

## 2.4   Discussions

The main advantage of MPNet over BERT and XLNet is that it conditions on more information while predicting a masked token, which leads to better learnt representations and less discrepancy with downstream tasks.

As shown in Table 1, we take a sentence $[\text{The}, \text{task}, \text{is}, \text{sentence}, \text{classification}]$ as an example to compare the condition information of MPNet/MLM (BERT)/PLM (XLNet). Suppose we mask two words $[\text{sentence}, \text{classification}]$ for prediction. As can be seen, while predicting a masked word, MP-

| Objective | # Tokens | # Positions |
|---|---|---|
| MLM (BERT) | 85% | 100% |
| PLM (XLNet) | 92.5% | 92.5% |
| MPNet | 92.5% | 100% |

Table 2: The percentage of input information (tokens and positions) leveraged in MLM, PLM and MPNet, assuming they predict the same amount (15%) of tokens.

Net conditions on all the position information to capture a global view of the sentence (e.g., the model knows that there two missing tokens to predict, which is helpful to predict two tokens "sentence classification" instead of three tokens "sentence pair classification"). Note that PLM does not have such kind of information. Furthermore, to predict a word, MPNet conditions on all preceding tokens including the masked and predicted ones for better context modeling (e.g., the model can better

predict "classification" given previous token "sentence", instead of predicting "answering" as if to predict "question answering"). In contrast, MLM does not condition on other masked tokens.

Based on the above example, we can derive Table 2, which shows how much conditional information is used on average to predict a masked token in each pre-training objective. We assume all the three objectives mask and predict the same amount of tokens (15%), following the common practice in BERT [2] and XLNet [5][3]. As can be seen, MLM conditions on 85% tokens and 100% positions since masked tokens contains position information; PLM conditions on all the 85% unmasked tokens and positions and $15\%/2 = 7.5\%$ masked tokens and positions[4], resulting in 92.5% tokens and positions in total. MPNet conditions on 92.5% tokens similar to PLM, and at the same time 100% positions like that in MLM thanks to the position compensation.

To summarize, MPNet utilizes the most information while predicting masked tokens. On the one hand, MPNet can learn better representations with more information as input; on the other hand, MPNet has the minimal discrepancy with downstream language understanding tasks since 100% token and position information of an input sentence is available to a model for those tasks (e.g., sentence classification tasks).

# 3 Experiments and Results

## 3.1 Experimental Setup

We conduct experiments under the BERT base setting ($\text{BERT}_{\text{BASE}}$) [2], where the model consists of 12 transformer layers, with 768 hidden size, 12 attention heads, and 110M model parameters in total. For the pre-training objective of MPNet, we randomly permute the sentence following PLM [5][5], choose the rightmost 15% tokens as the predicted tokens, and prepare mask tokens following the same 8:1:1 replacement strategy in BERT [2]. Additionally, we also apply whole word mask [12] and relative positional embedding [13][6] in our model pre-training since these tricks have been successfully validated in previous works [5, 14].

For pre-training corpus, we follow the data used in RoBERTa [7], which includes Wikipedia and BooksCorpus [15], OpenWebText [16], CC-News [17] and Stories [18], with 160GB data size in total. We use a sub-word dictionary with 30K BPE codes in BERT [2] to tokenize the sentences. Following the previous practice [17, 9], we limit the length of sentences in each batch as up to 512 tokens and use a batch size of 8192 sentences. We use Adam [19] with $\beta_1 = 0.9$, $\beta_2 = 0.98$ and $\epsilon = 1e-6$, and weight decay is set as 0.01. We pre-train our model for 500K steps to be comparable with state-of-the-art models like XLNet [5], RoBERTa [7] and ELECTRA [10]. We use 32 NVIDIA Tesla 32GB V100 GPUs, with FP16 for speedup. The total training procedure needs 35 days.

During fine-tuning, we do not use query stream in two-stream self-attention and use the original hidden states to extract context representations following [5]. The fine-tuning experiments on each downstream tasks are conducted 5 times and the median value is chosen as the final result. For experimental comparison, we mainly compare MPNet with previous state-of-the-art pre-trained models using the same $\text{BERT}_{\text{BASE}}$ setting unless otherwise stated. We also pre-train our MPNet in BERT large setting, where the weights are initialized by RoBERTa large model to save computations. The training details and experimental results on the large setting are attached in the supplemental material.

## 3.2 Results on GLUE Benchmark

The General Language Understanding Evaluation (GLUE) [20] is a collection of 9 natural language understanding tasks, which include two single-sentence tasks (CoLA [21], SST-2 [22]), three similarity and paraphrase tasks (MRPC [23], STS-B [24], QQP), four inference tasks (MNLI [25], QNLI [26], RTE [27], WNLI [28]). In our experiments, we will not evaluate WNLI to keep consis-

|  | MNLI | QNLI | QQP | RTE | SST | MRPC | CoLA | STS | Avg |
|---|---|---|---|---|---|---|---|---|---|
| *Single model on dev set* | | | | | | | | | |
| BERT [2] | 84.5 | 91.7 | 91.3 | 68.6 | 93.2 | 87.3 | 58.9 | 89.5 | 83.1 |
| XLNet [5] | 86.8 | 91.7 | 91.4 | 74.0 | 94.7 | 88.2 | 60.2 | 89.5 | 84.5 |
| RoBERTa [7] | 87.6 | 92.8 | **91.9** | 78.7 | 94.8 | 90.2 | 63.6 | **91.2** | 86.4 |
| MPNet | **88.5** | **93.3** | **91.9** | **85.8** | **95.5** | **91.8** | **65.0** | 91.1 | **87.9** |
| *Single model on test set* | | | | | | | | | |
| BERT [2] | 84.6 | 90.5 | 89.2 | 66.4 | 93.5 | 84.8 | 52.1 | 87.1 | 79.9 |
| ELECTRA [10] | **88.5** | **93.1** | 89.5 | 75.2 | **96.0** | 88.1 | **64.6** | **91.0** | 85.8 |
| MPNet | **88.5** | **93.1** | 89.9 | 81.0 | 96.0 | 89.1 | 64.0 | 90.7 | **86.5** |

Table 3: Comparisons between MPNet and the previous strong pre-trained models under $\text{BERT}_{\text{BASE}}$ setting on the dev and test set of GLUE tasks. We only list the results on each set that are available in the published papers. STS is reported by Pearman correlation, CoLA is reported by Matthew's correlation, and other tasks are reported by accuracy.

tency with previous works [2]. We follow RoBERTa hyper-parameters for single-task fine-tuning, where RTE, STS-B and MRPC are started from the MNLI fine-tuned model to be the consistent.

We list the results of MPNet and other existing strong baselines in Table 3. All of the listed results are reported in $\text{BERT}_{\text{BASE}}$ setting and from single model without any data augmentation for fair comparisons. On the dev set of GLUE tasks, MPNet outperforms BERT [2], XLNet [5] and RoBERTa [7] by 4.8, 3.4, 1.5 points on average. On the test set of GLUE tasks, MPNet outperforms ELECTRA [10], which achieved previous state-of-the-art accuracy on a variety of language understanding tasks, by 0.7 point on average, demonstrating the advantages of MPNet for language understanding.

### 3.3 Results on Question Answering (SQuAD)

| Method | SQuADv1.1 (dev) | SQuADv2.0 (dev) | SQuADv2.0 (test) |
|---|---|---|---|
| BERT [2] | 80.8/88.5 | 73.7/76.3 | 73.1/76.2 |
| XLNet [5] | 81.3/- | 80.2/- | -/- |
| RoBERTa [7] | 84.6/91.5 | 80.5/83.7 | -/- |
| MPNet | **86.9/92.7** | **82.7/85.7** | **82.8/85.8** |

Table 4: Comparison between MPNet and the previous strong pre-trained models under $\text{BERT}_{\text{BASE}}$ setting on the SQuAD v1.1 and v2.0. We only list the results on each set that are available in the published papers. The results are reported by exact match (EM)/F1 score.

The Stanford Question Answering Dataset (SQuAD) task requires to extract the answer span from the provided context based on the question. We evaluate our model on SQuAD v1.1 [26] dev set and SQuAD v2.0 [29] dev/test set (SQuAD v1.1 has closed its entry for the test set submission). SQuAD v1.1 always exists the corresponding answer for each question and the corresponding context, while some questions do not have the corresponding answer in SQuAD v2.0. For v1.1, we add a classification layer on the outputs from the pre-trained model to predict whether the token is a start or end position of the answer. For v2.0, we additionally add a binary classification layer to predict whether the answer exists.

The results of MPNet on SQuAD are reported on Table 4. All of the listed results are reported in $\text{BERT}_{\text{BASE}}$ setting and from single model without any data augmentation for fair comparisons. We notice that MPNet outperforms BERT, XLNet and RoBERTa by a large margin, both in SQuAD v1.1 and v2.0, which are consistent with the results on GLUE tasks, showing the advantages of MPNet.

## 3.4 Results on RACE

The ReAding Comprehension from Examinations (RACE) [30] is a large-scale dataset collected from the English examinations from middle and high school students. In RACE, each passage has multiple questions and each question has four choices. The task is to select the correct choice based on the given options.

The results on RACE task are listed in Table 5. We can only find the results from BERT and XLNet pre-trained on Wikipedia and BooksCorpus (16GB data) [7]. For a fair comparison, we also pre-train MPNet on 16GB data (marked as * in Table 5). MPNet greatly outperforms BERT and XLNet across the three metrics, demonstrating the advantages of MPNet. When pre-training MPNet with the default 160GB data, an additional 5.7 points gain (76.1 vs. 70.4) can be further achieved.

|  | | RACE | | IMDB |
|  | Acc. | Middle | High | Err. |
|---|---|---|---|---|
| BERT* | 65.0 | 71.7 | 62.3 | 5.4 |
| XLNet* | 66.8 | - | - | 4.9 |
| MPNet* | 70.4 | 76.8 | 67.7 | 4.8 |
| MPNet | **76.1** | **79.7** | **74.5** | **4.4** |

Table 5: Results on the RACE and IMDB test set under $\text{BERT}_{\text{BASE}}$ setting. "Middle" and "High" denote the accuracy on the middle school set and high school set in RACE. For IMDB, "*" represents pre-training only on Wikipedia and BooksCorpus (16GB size).

## 3.5 Results on IMDB

We further study MPNet on the IMDB text classification task [31], which contains over 50,000 movie reviews for binary sentiment classification. The results are reported in Table 5 [8]. It can be seen that MPNet trained on Wikipedia and BooksCorpus (16GB data) outperforms BERT and PLM (XLNet) by 0.6 and 0.1 point. When pre-training with 160GB data, MPNet achieves an additional 0.4 point gain.

| Model Setting | SQuADv1.1 | SQuADv2.0 | MNLI | SST-2 |
|---|---|---|---|---|
| MPNet | **85.0/91.4** | **80.5/83.3** | **86.2** | **94.0** |
| − position compensation (= PLM) | 83.0/89.9 | 78.5/81.0 | 85.6 | 93.4 |
| − permutation (= MLM + output dependency) | 84.1/90.6 | 79.2/81.8 | 85.7 | 93.5 |
| − permutation & output dependency (= MLM) | 82.0/89.5 | 76.8/79.8 | 85.6 | 93.3 |

Table 6: Ablation study of MPNet under $\text{BERT}_{\text{BASE}}$ setting on the dev set of SQuAD tasks (v1.1 and v2.0) and GLUE tasks (MNLI and SST-2). The experiments in ablation study are all pre-trained on the Wikipedia and BooksCorpus (16GB size) for 1M steps, with a batch size of 256 sentences, each sentence with up to 512 tokens.

## 3.6 Ablation Study

We further conduct ablation studies to analyze several design choices in MPNet, including introducing dependency among predicted tokens to MLM, introducing position compensation to PLM, etc. The results are shown in Table 6. We have several observations:

- After removing position compensation, MPNet degenerates to PLM, and its accuracy drops by 0.6-2.3 points in downstream tasks. This demonstrates the effectiveness of position compensation and the advantage of MPNet over PLM.

- After removing permutation operation but still keeping the dependency among the predicted tokens with two-stream attention (*i.e.*, MLM + output dependency), the accuracy drops slightly by 0.5-1.7 points. This verifies the gain of permutation used in MPNet.

- If removing both permutation and output dependency, MPNet degenerates to MLM, and its accuracy drops by 0.5-3.7 points, demonstrating the advantage of MPNet over MLM.

# 4 Conclusion

In this paper, we proposed MPNet, a novel pre-training method that addresses the problems of MLM in BERT and PLM in XLNet. MPNet leverages the dependency among the predicted tokens through permuted language modeling and makes the model to see auxiliary position information to reduce the discrepancy between pre-training and fine-tuning. Experiments on various tasks demonstrate that MPNet outperforms MLM and PLM, as well as previous strong pre-trained models such as BERT, XLNet, RoBERTa by a large margin. In the future, we plan to extend our MPNet into other advanced structures and apply MPNet on more diverse language understanding tasks.

## Acknowledgements and Disclosure of Funding

We thank the editor and each anonymous reviewer for their constructive comments and suggestions. This work was supported by National Key Research and Development Program of China under Grant 2017YFB1300205.

## Broader Impact

In natural language processing, pre-trained models have achieved rapid progress and a lot of pre-training methods are proposed. In terms of positive impacts, MPNet can help people to rethink current popular pre-trained methods (*i.e.*, MLM in BERT and PLM in XLNet) and thus inspire future researchers to explore more advanced pre-training methods. Currently, the pre-trained models have been considered as the most powerful tools, which help machine to obtain amazing performance in a series of NLP tasks. The final goal of the pre-trained models is to enable machine understands natural languages as human being, which is an important step towards artificial intelligence. In terms of negative impact, pre-training models are usually of large model size and training cost, which makes it too expensive for both research and product. In the future, we will develop more light-weight and low-cost pre-training models.

## Footnotes

[1]We do not consider next sentence prediction here since previous works [5, 7, 9] have achieved good results without next sentence prediction.

[2]One trivial solution is to let the model see all the input tokens, i.e., $115\% * n$ tokens, but introduces new discrepancy as the model can only see $100\% * n$ tokens during fine-tuning.

[3]XLNet masks and predicts 1/7 tokens, which are close to 15% predicted tokens.

[4]PLM predicts the $i$-th token given the previous $i-1$ tokens autoregressively. Therefore, the number of tokens can be leveraged on average is $(n-1)/2$ where $n$ is the number of predicted tokens.

[5]Note that we only improve upon PLM in XLNet, and we do not use long-term memory in XLNet.

[6]We follow [14] to adopt a shared relative position embedding across each layer for efficiency.

[7]The results of BERT are from the RACE leaderboard (`http://www.qizhexie.com/data/RACE_leaderboard.html`) and the results of XLNet are obtained from the original paper [5].

[8]The result of BERT is from [32] and the result of XLNet is ran by ourselves with only PLM pre-training objective in Transformer.

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
