[Supplementary Material · appendix.pdf]

# A  Pre-training Hyper-parameters

The pre-training hyper-parameters are reported in Table 7. For $BERT_{LARGE}$, our model is initialized by $RoBERTa_{LARGE}$ to save computations, and also disable relative position embedding and whole word mask to keep consistent with RoBERTa setting. We also apply a smaller learning rate of 5e-5 for continual training since it has been well optimized.

| Hyper-parameter | Base | Large |
|---|---|---|
| Number of Layers | 12 | 24 |
| Hidden Size | 768 | 1024 |
| Filter Size | 3072 | 4096 |
| Attention heads | 12 | 16 |
| Dropout | 0.1 | 0.1 |
| Weight Decay | 0.01 | 0.01 |
| Learning Rate | 6e-4 | 5e-5 |
| Steps | 500K | 100K |

Table 7: Pre-training hyper-parameters for $BERT_{BASE}$ and $BERT_{LARGE}$ setting.

# B  Fine-tuning Hyper-parameters

The fine-tuning hyper-parameters are reported in Table 8.

| Hyper-parameter | RACE | SQuAD | GLUE |
|---|---|---|---|
| Learning Rate | 1.5e-5 | 3e-5 | 1e-5,2e-5,3e-5 |
| Batch Size | 16 | 48 | 16, 32 |
| Weight Decay | 0.1 | 0.01 | 0.1 |
| Epochs | 5 | 4 | 10, 15 |
| Learning Rate Decay | Linear | Linear | Linear |
| Warmup Ratio | 0.06 | 0.06 | 0.06 |

Table 8: Fine-tuning hyper-parameters for RACE, SQuAD and GLUE.

# C  MPNet on Large Setting

We pre-train our MPNet on large setting with a initialization of $RoBERTa_{LARGE}$ to save computation. The results of MPNet on $BERT_{LARGE}$ setting are reported in Table 9. From Table 9, we observe that our MPNet outperforms RoBERTa [7] by 0.5 points on average. Since it is only pre-trained 100K steps for a quick verification, which cannot fully demonstrate the advantages of our method. Therefore, we are also preparing the large-level model of MPNet from scratch, and will update it when it is done.

| | MNLI | QNLI | QQP | RTE | SST | MRPC | CoLA | STS | Avg |
|---|---|---|---|---|---|---|---|---|---|
| RoBERTa [7] | 90.2 | 94.7 | 92.2 | 86.6 | 96.4 | 90.9 | 68.0 | 92.4 | 88.9 |
| MPNet | **90.5** | **94.9** | 92.2 | **88.0** | **96.7** | **91.4** | **69.1** | 92.4 | **89.4** |

Table 9: The results of MPNet on $BERT_{LARGE}$ setting on GLUE development set.

# D  Effect of MNLI initialization

In our paper, we adopt MNLI-initialization for RTE, STS-B and MRPC to be consistent with the fine-tuning setting of RoBERTa. To make a fair comparison, we also carry experiments on RTE,

| Model | RTE | STS-B | MRPC | GLUE |
|---|---|---|---|---|
| MPNet | 81.0 | 90.7 | 89.1 | 86.5 |
| – MNLI-init | 79.8 | 90.7 | 88.7 | 86.3 |

Table 10: Results of RTE, STS-B and MRPC on the test set without MNLI-initialization. "- MNLI-init" means disabling MNLI-initialization in MPNet. "GLUE" means the average score on all GLUE tasks.

STS-B, and MRPC without MNLI-initialization to analyze the impact of MNLI-initialization. The results are shown in Table 10. From Table 10, we observe that removing MNLI-initialization only slightly hurts the performance on RTE and MRPC, but still outperforms ELECTRA on average score.

# E    Training Efficiency

To further demonstrate the effectiveness of our method, we also investigate the training efficiency of our MPNet in relative to other advanced approaches. The comparisons are shown in Table 11. When compared to XLNet/RoBERTa, we found our method can achieve better performance with fewer computations.

| Model | Train FLOPS | GLUE |
|---|---|---|
| BERT | 6.4e19 (0.06×) | 83.1 |
| XLNet | 1.3e21 (1.10×) | 84.5 |
| RoBERTa | 1.1e21 (0.92×) | 86.4 |
| MPNet-300K | 7.1e20 (0.60×) | 87.7 |
| MPNet-500K | 1.2e21 (1.00×) | 87.9 |

Table 11: Comparisons of training flops in different methods under $\text{BERT}_{\text{BASE}}$ setting. BERT is trained on 16GB data and others are on 160GB data.

# F    More Ablation Studies

We further conduct more experiments to analyze the effect of whole word mask and relative positional embedding in $\text{BERT}_{\text{BASE}}$ setting. The results are reported in Table 12, which are also pre-trained on the Wikipedia and BooksCorpus (16GB size) for 1M steps, with a batch size of 256 sentences and each sentence with up to 512 tokens.

| Model Setting | SQuAD v1.1 | | SQuAD v2.0 | | GLUE | |
|---|---|---|---|---|---|---|
| | EM | F1 | EM | F1 | MNLI | SST-2 |
| MPNet | **85.0** | **91.4** | **80.5** | **83.3** | **86.2** | **94.0** |
| – whole word mask | 84.0 | 90.5 | 79.9 | 82.5 | 85.6 | 93.8 |
| – relative positional embedding | 84.0 | 90.3 | 79.5 | 82.2 | 85.3 | 93.6 |

Table 12: Ablation study of MPNet under $\text{BERT}_{\text{BASE}}$ setting on the dev set of SQuAD tasks (v1.1 and v2.0) and GLUE tasks (MNLI and SST-2).

# G    Training Speedup

We introduce the training tricks to speed up the training process by partitioning the attention matrix. As shown in Figure 4, we divide the attention matrix (originally shown in Figure 2 in the main paper) into 4 sub-matrices according to that the query and the key/value are from the non-predicted part or the predicted part. More specifically, the query of matrix-{A,B} is from the non-predicted part,

the query of matrix-{C,D} is from the predicted part, the key/value of matrix-{A,C} is from the non-predicted part, the key/value of matrix-{B,D} is from the predicted part. We find that matrix B has no use for our model training. Therefore, we only consider matrix-{A, C, D} when computing the content stream, which will save nearly 10% computations during the whole training process.

Figure 4: The attention mask matrix for the content stream of MPNet. More details can refer to Figure 2 in the main paper.