[Reviews · NeurIPS 2020]

Review 1

Summary and Contributions: Masked Language Modelling (MLM) ignores the dependency between masked tokens when predicting them. Permuted language modelling (PLM) captures this dependency, but introduces a discrepancy between pre-training and fine-tuning. This work aims to achieve the best of both worlds by including the masked tokens and their positions in the dependency of the predicted tokens, resulting in improved pre-training performance compared to BERT (MLM) and XLNET (PLM).

Strengths: The idea presented here is reasonably simple, but required interesting modifications to existing pre-training setups, and is explained well with nice diagrams. The results are good across many standard datasets (GLUE benchmark, Squad etc.), and the ablation study shows the benefits of the method clearly. This is a popular and high-impact area, so it would be useful for the community to be exposed to these results.

Weaknesses: There are some misleading comparisons to ELECTRA. The method is a bit incremental on top of BERT/XLNET. Update: My concerns re ELECTRA were adressed.

Correctness: As far as I can tell.

Clarity: Mostly yes.

Relation to Prior Work: Mostly yes. Alternative approaches, particularly ELECTRA, could be better explained.

Reproducibility: Yes

Additional Feedback: When comparing to ELECTRA-base, you should not use RTE, STS-B and MRPC scores when initialised from the MNLI fine-tuned model for a fair comparison, these experiments should be easy to run. It's important that the it's clear to the reader where your improvements come from, whether from your pre-training method, or MNLI initialisation. I would also briefly explain the ELECTRA method. I would reference the ablation study in the appendix somewhere in the main paper. For clarity I would suggest including pseudo code or toy pytorch code describing your attention mechanism and masking strategy, but this is optional! typos: Section '2.4 Advantage' should have a clearer name. l138 'the the' If possible you could have your submission proof-read for English style and grammar issues. Update: Given my concerns were adressed, I've increased my score.


Review 2

Summary and Contributions: In this paper, the authors propose a new pre-trained language model called MPNet, which combines the advantages of both BERT (MLM) and XLNet (PLM). The proposed MPNet leverages the dependency among predicted tokens through PLM and takes auxiliary position information as input to reduce the position discrepancy. In practice, to combine MLM and PLM poses great challenge, and the authors propose several delicate designs to overcome the issues. The experiments are mainly carried out on a base model (i.e., 110M params), and the results show that the proposed MPNet could give consistent and significant improvements over similar baseline models. Overall, the idea of the paper is straightforward and easy to understand, which is a natural extension to combine the MLM and PLM. The main concern of this paper is that most of the results are on a base-level model, which may diminish the impact of the results. This is because most of the main-stream works on pre-trained language models (such as BERT, XLNet, ALBERT, ELECTRA) mainly report their results on a large transformer model (such as 24-layer). As NeurIPS is one of the highest publications in this field, it is recommended to adjust the content to reflect this issue in the next version.

Strengths: 1) A natural and straightforward idea to combine two popular pre-training tasks (MLM and PLM). 2) Experimental results on a base-level model are significantly better than previous works.

Weaknesses: 1) The main results are on a base-level model, which may weaken the experimental impact of this paper.

Correctness: The proposed methods are technically sound.

Clarity: The content of this paper is clearly written.

Relation to Prior Work: This work primarily combines and extends two popular pre-trained language models: BERT and XLNet. The authors have discussed the relations and differences between theirs and previous works.

Reproducibility: Yes

Additional Feedback: Questions: 1) The main results are on a base-level model, and some results are on a large-level model (in appendix). Why didn't you put the results on a large-level model in the main paper? If the time matters, then I would suggest the authors to do so in the next version, as I think most of the readers would like to see the performance your model on a larger model setting. 2) The development scores are chosen as 5-time median values. How about the test scores? As GLUE and SQuAD require to submit either prediction file or original source codes to get test scores, did you submit your best-performing model on dev set? Minor comments: 1) In line 19, suggestion: 'Pre-training models' -> 'Pre-trained language models' 2) In line 175-181, the right double quote seems weird compared to the one in the caption of Table 1 (seems to be correct). 3) In line 208, the original optimizer in BERT is Adam with weight decay optimizer, but you did not indicate 'weight decay' here. 4) In line 229, 'GELU' -> 'GLUE' After response: Thank you for your response. I remain my original score and vote for weak accept.


Review 3

Summary and Contributions: The paper proposes a new pretraining objective by combining ideas from masked language modeling (BERT) and permuted language modeling (XLNet). The objective is that of XLNet, which is an expectation over prediction order of missing tokens of predicting the tokens, augmented with positional embeddings of future, yet to be predicted, tokens (the latter is available to BERT but not XLNet). The advantages presented are that this both introduces dependency in the predicted tokens, and has information about future tokens (the positions/count) which is closer to the fine tuning objectives. The model has evaluation with BERT, XLNet, RoBERTa as benchmarks, on GLUE, SQuAD, RACE, IMDB. Also, an ablation study in a smaller data regime shows that permutation/MLM/output dependency are all useful.

Strengths: - Evaluation is reasonably convincing. - Ablation studies show that the model features are useful (in a smaller data regime).

Weaknesses: - The paper doesn't discuss the comparison of the new objective compared to standard language modeling. - The new model is a bit incremental but seems effective based on empirical evidence.

Correctness: The experiments seem to support the claims. The ablation study on smaller data supports the objective features. The other experiments use pre-trained models from previous work and so it is a bit unclear if the two setups are similar enough, ie would have been nice, but computationally expensive, if the authors trained XLNet in their code base.

Clarity: Yes.

Relation to Prior Work: Yes.

Reproducibility: Yes

Additional Feedback:


Review 4

Summary and Contributions: This paper proposes a new pre-training method called MPNet that based on BERT and XLNet and inherits their advantages and avoids their limitations. The author combines BERT and XLNet in a unified view, that the non-masked tokens are put on the left side while the masked and to be predicted tokens are on the right side of the permuted sequence for both MLM and PLM. With this combination, MPNet can inherit Output Dependency from PLM and Input Consistency from MLM. The author conducts experiments under the BERT base setting in GLUE, Squad, RACE, IMDB, and showing the advantages of MPNet.

Strengths: The idea is fairly simple, just as other influential ideas in this domain, such as BERT and XLNet. The methodology is also simple to implement. In this empirical area of research, this paper has tried running on multiple benchmarks and compare with many baseline models.

Weaknesses: All the results reported in the paper are based on BASE models. I found the LARGE results in the supplemental material, however, the improvements are very marginal then, e.g., no improvement on QQP, STS, and less than 0.3% MNLI, QNLI, SST. In contrast, the advances are quite substantial on BASE settings, as shown in Tables 3-4. Therefore, the improvement inconsistence puts the proposed model in a doubtful position if it really works in a general way. It is expected to see more experiments on BERT large setting to illustrate the effectiveness of MPNet also on strong baselines, or at least, some insightful discussion is needed to explain such weaker results. Besides, some adopted datasets like RTE and MRPC training sets are too small to give persuasive results. It is recommended that SuperGLUE is used for further comparison of model performance.

Correctness: yes

Clarity: well written

Relation to Prior Work: Few recent research about pre-trained language modeling methods and the SOTA methods in the downstream tasks (however maybe those approaches are not the same to this work) were discussed. It would be better to have a related work section, i.e., to discuss the recent advances of language representations and significance of this work.

Reproducibility: Yes

Additional Feedback: Since the proposed method would be general, why not use the SOTA models as backbone, e.g., ALBERT, to see if the method can be applied in a much stronger backbone? What about the training efficiency, e.g., training speed, GPU cost, etc., compared with BERT and XLNet?

[Author Response · NeurIPS 2020]

Thank all reviewers for the valuable comments and suggestions. Please find responses (R) to specific comments (C).

**To Reviewer #1**

C1: *Misleading comparisons to ELECTRA in RTE, STS-B and MRPC.*

R1: In our submission version, we adopt MNLI-initialization for RTE, STS-B, and MRPC to be consistent with
the fine-tuning setting of RoBERTa. Actually, we have carried experiments on RTE, STS-B, and MRPC without
MNLI-initialization to make a fair comparison with ELECTRA. The results are shown in Table 1. Removing MNLI-
initialization only slightly hurts the performance on RTE and MRPC, but still outperforms ELECTRA on average score.
We will add this comparison into our paper in the next version.

C2: *Other suggestions and minor typos.*

R2: Thanks for your suggestions. We will refer the ablation study in the appendix somewhere in the main paper, add
pseudo code to describe our attention mechanism and mask strategy, and also fix the typos in the later version.

**To Reviewer #2**

C1: *The results are mainly reported on a base-level model.*

R1: Pre-training on large-level model requires huge amount of computation resource. Therefore, in this version,
we initialize our large-level model from the RoBERTa model and continue to pre-train only 100K steps for a quick
verification, which cannot fully demonstrate the advantages of our method. We are pre-training the large-level model
from scratch, and will update the results when it is finished.

C2: *Do you submit test scores based on the best performing model on dev set.*

R2: Yes. Following the previous practice [1], we submit the best performing model on the dev set to evaluate the test
scores.

[1] BERT: Pre-training of Deep Bidirectional Transformers for Language Understanding.

C3: *Minor issues.*

R3: Thanks for your valuable suggestions. We will refine these issues in the later version.

**To Reviewer #3**

C1: *Providing a discussion about the comparisons between the proposed method and standard language modelings.*

R1: Thanks for the suggestion. As mentioned in Section 2.3-2.4, we have discussed the comparisons between MPNet
and masked/permuted language modeling. We will add the discussion about our method and standard language modeling
in the later version.

**To Reviewer #4**

C1: *Improvements in the large-level model.*

R1: Pre-training on large-level model requires huge amount of computation resource. Therefore, in this version, we
initialize our large-level model from the pre-trained RoBERTa model and continue to pre-train only 100K steps for a
quick verification, which cannot fully demonstrate the advantages of our method. We are pre-training the large-level
model from scratch, and will update the results when it is finished.

C2: *SuperGLUE is recommended.*

R2: Thanks for your recommendation. We will prepare experiments on SuperGLUE and report it in the later version.

C3: *It will be better to have a discussion about the recent advances of researches and significance of this work.*

R3: Thanks for your advice. We will add a related work section to discuss the recent research about pre-trained language
modelings to analyze the recent advances and significance of our work.

C4: *Why not choose the SOTA methods as backbone (e.g., ALBERT)?*

R4: We adopt BERT based structure as our backbone since it is one of the most popular architectures used in this field.
We will conduct experiments based on other SOTA models (e.g., ALBERT) to manifest the generality of our method in
the future.

C5: *What about training efficiency?*

R5: The training efficiency are reported in Table 2. When compared to XLNet/RoBERTa, we found our method can
achieve better performance, but fewer computations. We will add the comparisons of training efficiency in the new
version.

| Model | RTE | STS-B | MRPC | GLUE |
|---|---|---|---|---|
| ELECTRA | 75.2 | 91.0 | 88.1 | 85.8 |
| MPNet | 81.0 | 90.7 | 89.1 | 86.5 |
| - MNLI-init | 79.8 | 90.7 | 88.7 | 86.3 |

Table 1: Results of RTE, STS-B and MRPC on the test set without MNLI-initialization. "- MNLI-init" means disabling MNLI-initialization in MPNet. "GLUE" means the average score on all GLUE tasks.

| Model | Train FLOPS | GLUE |
|---|---|---|
| BERT | 6.4e19 (0.06×) | 83.1 |
| XLNet | 1.3e21 (1.10×) | 84.5 |
| RoBERTa | 1.1e21 (0.92×) | 86.4 |
| MPNet-300K | 7.1e20 (0.60×) | 87.7 |
| MPNet-500K | 1.2e21 (1.00×) | 87.9 |

Table 2: Comparisons of training flops in different methods under $\text{BERT}_{\text{BASE}}$ setting. BERT is trained on 16GB data and others are on 160GB data.

[Meta-Review · NeurIPS 2020]

This paper proposes a new approach to self-supervised pretraining on text, building very closely on prior work in BERT and XLNet. The paper demonstrates that the approach yields models that are noticably better than comparable models from prior work, and isolates the reason for this in a reasonably thorough ablation. While one reviewer raises concerns about the quality of comparisons, I'm convinced that they are already pretty much up to the standards of the field, and will be fully satisfactory after the promised revisions/additions. While this work is perhaps somewhat incremental, the method is effective and relevant to a highly prominent open problem in ML for text.